# Position-Aware Indoor Human Activity Recognition Using Multisensors Embedded in Smartphones

**DOI:** 10.3390/s24113367

**Published:** 2024-05-24

**Authors:** Xiaoqing Wang, Yue Wang, Jiaxuan Wu

**Affiliations:** 1School of Information Science and Engineering, Shenyang Ligong University, Shenyang 110168, China; wang_xq@sylu.edu.cn; 2School of Computer Science and Technology, Anhui University of Technology, Maanshan 243099, China; yuewang@ahut.edu.cn

**Keywords:** human activity recognition, indoor positioning, smartphone sensors

## Abstract

Composite indoor human activity recognition is very important in elderly health monitoring and is more difficult than identifying individual human movements. This article proposes a sensor-based human indoor activity recognition method that integrates indoor positioning. Convolutional neural networks are used to extract spatial information contained in geomagnetic sensors and ambient light sensors, while transform encoders are used to extract temporal motion features collected by gyroscopes and accelerometers. We established an indoor activity recognition model with a multimodal feature fusion structure. In order to explore the possibility of using only smartphones to complete the above tasks, we collected and established a multisensor indoor activity dataset. Extensive experiments verified the effectiveness of the proposed method. Compared with algorithms that do not consider the location information, our method has a 13.65% improvement in recognition accuracy.

## 1. Introduction

Human activity recognition (HAR) is of great help in elderly care and rehabilitation [1]. Although various deep learning models have been successfully applied to recognize human activities, most existing research has focused on simple actions such as walking, standing, and running, which are typically characterized by actions or postures. In contrast, compound indoor activities such as eating and using the bathroom can better reflect people’s daily lives and are more useful for elderly care but are less studied. In this research, we focused on elderly care as a potential application and propose an indoor activity recognition method based on smartphones. We chose to use smartphones to collect data because, currently, smartphones generally have built-in sensors such as gyroscopes, accelerometers, and magnetometers, which can be conveniently used to collect data without the need for participants to purchase and carry additional equipment [2]. Using smartphones as the data acquisition equipment would be more user-friendly, and the feasibility of using mobile phone sensor data to recognize simple human actions such as “Sit” and “Walk” has been widely studied and proven [3]. However, due to the higher level of semantics of compound activities such as “Eating” and “Using bathroom”, identifying them not only requires movement features, but also contextual information of the surrounding environment. Therefore, compared to simple action recognition, compound activity recognition is more challenging. In addition, designing effective sensor solutions for compound activities is also a challenging task.

The most commonly used sensors in human action recognition are accelerometers and gyroscopes, the data features of which mainly reflect the motion characteristics of different movements, such as walking, sitting, standing, lying down, etc. Researchers collect time series signals from these sensors, and feature vectors of different motions are then extracted from the raw data using either traditional machine learning methods [4] or newly developed deep learning methods [1] to finally train an classifier. However, these sensor signals are difficult to further distinguish between specific household activities, such as working at a desk and eating, both of which use sitting movements, while cooking and doing laundry usually use postures of standing and bending down. For these indoor home activities, in addition to the strong correlation between the activity types and the aforementioned motion characteristics, there is also a close correlation between indoor locations and activities [5]. For example, cooking can only happen in the kitchen, while doing laundry is more likely to occur in the bathroom. Therefore, we propose to incorporate indoor location information into indoor HAR. Compared with indoor positioning technologies such as UWB and Bluetooth, geomagnetic positioning technology can rely solely on the built-in sensors of smartphones to achieve positioning, without the need to deploy additional infrastructure in the environment in advance [6]. This can not only reduce deployment costs but also increase the practicality of the system. Although many scholars have studied the feasibility of using geomagnetic sensors for localization [7], the focus of these methods is on precise positioning rather than identifying human activities, so they need to establish an accurate geomagnetic map in advance and limit the collection direction of geomagnetic sensors to ensure measurement accuracy, which is inconvenient for the task of using built-in sensors in mobile phones to assist in identifying human activities. In this study, we designed and integrated a simplified positioning method into the indoor activity recognition system, using neural networks to extract geographical features contained in the geomagnetic data, estimating the type of room rather than accurately locating user coordinates. In addition, inspired by the combination of optical information and Bluetooth technology in [8] for indoor positioning, we collected environmental light information detected by the built-in optical sensors of mobile phones and combined it with geomagnetic information to jointly complete indoor positioning.

To validate and test the method proposed in this article, we collected and established a smartphone sensor dataset for indoor activity recognition using four different sensors, including geomagnetic sensors, gyroscopes, acceleration sensors, and optical sensors. We recorded six common types of indoor activities that occurred in five different indoor rooms. In order to better utilize these multisensor data, we developed an indoor activity recognition model that takes both the location and motion features into consideration and we refer to our model as LM-IAR. Our methodology includes a location feature extraction subnetwork (LFN) using data from a geomagnetic sensor and an optical sensor, a motion feature extraction subnetwork (MFN) using gyroscope and accelerometer signals, and a feature fusion layer that fuses multimodal location features and action features.

The main contributions of this study are as follows:We propose to combine indoor positioning technology with motion recognition using the built-in sensors of mobile phones to achieve compound indoor human activity recognition, which can increase the spatial and temporal correlation between activity types and lifestyle habits;We propose a method for extracting indoor location features based on geomagnetic and ambient light data using convolutional neural networks. Our method does not require the subject’s phone posture to always be consistent, which greatly improves user friendliness;We collected and built a dataset of multimodal sensors embedded in smartphones for compound indoor activity recognition, including sleeping, eating, doing laundry, using the bathroom, and desk work. Based on this dataset, we trained a deep learning model with a multimodal data fusion structure.

## 2. Related Work

There are two mainstream data collection methods for human activity recognition: vision-based [9] and sensor-based [10] methods. Vision-based methods collect and analyze images or videos through cameras to identify the types of human activities [11]. This kind of method does not require the subject to carry any equipment. However, issues such as the limited sensing distance and the indirect angle between the camera and the captured object can seriously affect the accuracy. The sensor-based method utilizes sensors placed on the human body or in the environment to estimate human movement [12]. In order to better capture motion details, some sensor-based methods require participants to wear wearable devices to collect data. For example, Mathangi Sridharan et al. [13] identified behaviors of patients through wearable sensors, and P. Kalyani et al. [14] also obtained data through wearable devices to identify activities. However, wearing additional wearable sensors on the human body may make users feel uncomfortable in their daily lives. Due to the development of the Internet of Things and intelligent technology, various types of sensors have been widely embedded in portable devices such as mobile phones, watches, etc. Rafik Djemili et al. [15] proposed a deep-learning-based human activity recognition method using inertial sensors built into smart phones. Compared to data collected using wearable devices, data collected by mobile phones often have greater noise interference, resulting in lower accuracy, but being more user-friendly.

Classification algorithms for activity recognition are being widely studied. As early as 2007, Bao et al. [16] began research on human behavior recognition. They tied five biaxial accelerometers to the buttocks, wrists, arms, knees, and thighs of subjects, extracted their basic features, and then used various methods such as K-nearest neighbors and decision trees to classify and recognize 20 types of activities. Murad et al. [17] used naive Bayesian and K-nearest neighbor algorithms to explore the ability of integrating built-in accelerometers and gyroscopes in smartphones to recognize human body activities. These traditional machine learning methods require determining classification methods based on requirements; the modeling process is complex and the model is inflexible. With the development of deep learning, end-to-end activity recognition methods have gradually emerged. Zeng et al. [18] used convolutional neural networks to convolve accelerometer data. Hammerla et al. [19] used long short-term memory networks, bidirectional long short-term memory networks, and convolutional neural networks for behavior recognition. Xia et al. [20] proposed an LSTM-CNN model for HAR. Shavit et al. [21] used transformer models for human behavior recognition based on inertial sensors. Sekaran et al. [22] proposed a lightweight smartphone-based multihead convolutional network for human activity recognition. In these studies, the behaviors that can be recognized are relatively simple, such as running, walking, standing, and lying down, while more complex activities such as working and eating are hard to distinguish. Due to the inseparable relationship between family activities and location, we decided to consider indoor location when identifying indoor activities.

In the field of indoor positioning, researchers have explored various environmental signals used for positioning, such as WiFi, Bluetooth, visible light, etc. However, WiFi [23], Bluetooth [24], and visible-light-based [25] methods require the deployment of additional infrastructure, which is inconsistent with the user-friendly indoor activity recognition method pursued in this article. Compared with the above methods, indoor geomagnetism can achieve positioning without the need for special infrastructure deployment. Some researchers have considered using the spatial features of geomagnetic signals for localization. For example, SemanticSLAM [26] proposed collecting geomagnetic signals from landmark locations in advance to determine the current position. However, the characteristics of different points may be similar, resulting in limited spatial discrimination of signals from these scattered points. Other researchers propose to improve localization accuracy by comparing the features of multiple continuous signals. For example, Travi Navi [27] take signal sequences as inputs. However, comparing two sequences, especially when using relatively long signal sequences as inputs, may result in higher computational costs. Inspired by the success of deep learning algorithms, some methods have emerged that utilize neural networks to process temporal data and predict positions. For example, the work in [28] proposed to use recurrent neural networks (RNNs) to extract temporal sequence features for position localization. DeepML [29] designed a indoor positioning system based on long short-term memory (LSTM). ST-loc [7] proposed to use Resnet and bidirectional LSTM to extract features from geomagnetic data. Shu et al. [30] proposed a direction-aware multiscale recurrent neural network for geomagnetic positioning. However, these methods were all proposed for more accurate positioning and require detailed collection of geomagnetic fingerprints during the offline phase.

## 3. Data Acquisition

To train the indoor location and activity recognition model, we first established a smartphone multimodal sensor dataset containing activity tags as well as location tags. In this section, we will detail the design, collection, and processing methods of the dataset.

### 3.1. Types and Locations of Indoor Activities

A typical apartment is chosen as the indoor activity data collection site, as shown in Figure 1. The total area of the apartment is 132 square meters, including two bedrooms, a kitchen, a bathroom, a living room, and a dining room:Eating: In the dining room;Sleeping: In bedroom 1 or 2;Using bathroom: In the bathroom;Doing laundry: In the bathroom;Cooking: In the kitchen;Desk work: In the living room or bedroom 2.

We invited volunteers to install data collection software called Sensor Logger Version 1.15.2 on their mobile phones, and set the collection frequency as 100 Hz. Volunteers were instructed to hold their phones with their hands while simulating the six types of activities mentioned above. When holding the phone becomes inconvenient, they should keep it nearby. For example, when collecting activities such as eating and desk work, they hold their phones while walking to the table and sitting down. After sitting down, they placed their phones on the table. After collecting the data, we cut off the initial 3 s and final 3 s of each sample segment, as these data usually contain interference.

Figure 2 illustrates the data collection results for various activities. Typically, an activity data sequence comprises both handheld time period data and stationary time period data. As shown in the figure, when the phone is held by hand, gyroscopes and accelerometers can capture richer motion information, such as standing, walking, sitting, lying down, etc. When the phone is placed nearby, the data from inertial sensors usually remain relatively stable, whereas the collected ambient light and geomagnetic information will be more accurate. As shown in the figure, different activities may involve similar combinations of actions, such as eating and desk work both involving walking to the table, sitting down, placing the phone down on the table, etc. Therefore, relying solely on motion features may lead to misidentification, and in this case, auxiliary position information will be helpful.

### 3.2. Data Collection Results

We used the Sensor Logger APP on Android system for raw data collection and collected data from four sensors that are currently widely embedded in smartphones: magnetometer, gyroscope, accelerometer, and optical sensor. When volunteers simulated the aforementioned six household activities in corresponding rooms, sets of sequential data were being collected from the sensors. More specifically, a collected set of date sequence *S* consists of the following signals:(1)S={B,A,G,L}
where $B=\{B_{x},B_{y},B_{z}\}$ denotes three-axis geomagnetic intensity data, $A=\{A_{x},A_{y},A_{z}\}$ denotes three-axis acceleration data, $G=\{G_{x},G_{y},G_{z}\}$ denotes three-axis angular velocity data, and *L* refers to one-axis ambient light intensity data.

In addition, we also invited volunteers to walk around each room in daylight and at night, and collected more geomagnetic data and ambient light data to better train the position recognition network. The data collection time is about 20 s each time, and the sampling frequency is 100 Hz. The total number of sets of data sequences collected for various activities and rooms is shown in Table 1.

The collected data inevitably introduce some noise. Therefore, we apply a weighted recursive average filter to the raw signal. We use a sliding window with a fixed length *N*. Each time new data arrive, we remove the oldest data from the window queue and add the new data. Then, we calculate the weighted average of all data in the current window and output it. This output value becomes the filtered signal output.
(2)$$A_{k}=\left(1-\alpha \right)A_{k-1}+\alpha D_{k}$$
where $D_{k}$ is the new data, $A_{k}$ is the filtered output, $A_{k-1}$ is the output of last time, and α=2/(N+1) is the weight.

In order to facilitate subsequent model training, we further normalize and segment the original data sequence mentioned above. We normalize the data to range of [0,1] before inputting them to our recognition model:(3)$$X_{i}=\frac{x_{i}-x_{min}}{x_{max}-x_{min}}\left(i=1,2,\ldots n\right)$$
where $x_{i}$ is the original sensor data, *n* is the number of the sensor data sequence, $x_{min},x_{max}$ are the minimum and maximum data of the sequence, and $X_{i}$ is the normalized data.

The length of each segmented data is 1024, and the overlap step is 200. The details of the training and testing dataset used in the experiment will be described in the experiment section.

## 4. Proposed Method

### 4.1. Overall Structure of Proposed Method

The occurrence of indoor activities is closely related to their locations, so our model combines indoor positioning with indoor activity recognition, extracts location and motion features of indoor activities from different sensors, and fuses them to ultimately achieve activity recognition. The overall structure of the proposed method LM-IAR is shown in Figure 3. We divide the collected sensor data into two parts for location and motion feature extractions separately. The three-axis geomagnetic data and ambient light data are sent to the indoor location feature extraction subnetwork (LFN), which outputs location information features. At the same time, data collected from gyroscope and accelerometer sensors are sent to the motion feature extraction subnetwork (MFN) to extract motional features of activities. Then, the feature fusion layer (MFL) fuses these multimodal features and sends them into the final indoor activity recognition subnetwork (IAN) to identify the activity type. The details of each part of our model will be described in Section 4.2, Section 4.3 and Section 4.4.

Among the four sensors we use, the data collected by gyroscopes and accelerometers are used to extract motion features, which are time-related. Therefore, we directly use the collected time series data and design a transformer encoder structure network to extract action features. The location feature of geomagnetic and ambient light data are extracted using CNN networks. We choose to use CNN for two reasons. Firstly, we are more concerned with the spatial distribution of these data rather than their temporal characteristics. Secondly, for the sake of user friendliness, we did not require the subjects to maintain stable sensor postures like other geomagnetic positioning methods. Therefore, the geomagnetic data are inevitably affected by the measurement direction. Therefore, we convert the three-axis geomagnetic data into a two-dimensional matrix and use the convolutional process of the 2D-CNN network to reduce this impact.

### 4.2. Location Feature Extraction

Geomagnetic positioning technology utilizes the different distribution characteristics of geomagnetic field intensity with geographic location changes to achieve the positioning of moving carriers. Figure 4 shows the geomagnetic data collected from different rooms.

As mentioned earlier, in this study, we did not pursue accurate coordinate localization, but treated localization as a classification problem with the aim of extracting features of the room where the subjects were located. Location features are extracted from geomagnetic $\left\{B_{x},B_{y},B_{z}\right\}$ and light intensity data *L*. From a user-friendly perspective, we do not require users to always maintain consistent phone directions during the data collection process, but the direction of the phone will inevitably have an impact on geomagnetic sequences. Therefore, instead of directly using temporal sequences, we convert the three-axis temporal geomagnetic data into a three-channel image and use a two-dimensional CNN to extract positional features from the geomagnetic signal. In addition, considering the different lighting conditions in different rooms, we introduce ambient light information as the fourth input channel into the network to enhanced the indoor location classification. We divide the data sequences into sequence segments with the length of 1024, normalize them to a range of 0–255, and then convert and combine the data into arrays with the size of 32×32×4, as shown in Figure 5, and then use a modified Resnet [31] to extract its feature $Z_{l}$.

Given the limited amount of data we collect, performing transfer learning based on parameters pretrained on the ImageNet dataset is a common approach. However, this pretrained model is trained based on natural images, which is very different from the images converted from time series used in our work. Transfer learning research [32] has shown that when the source domain and target domain are far apart, adding FC layers to the pretrained during transfer learning can be helpful. Therefore, we choose ResNet18 as the basic structure, remove the original classification layer, and replace it with two fully connected layers. As shown in Figure 6, the outputs of the two FC layers are 128 and 6 dimensions, respectively. During the transfer learning process, we use softmax at the end of the model to classify the input data into one of the six room classes. After fine-tuning the parameters, we remove the final fully connected layer and use 128-dimensional features $Z_{l}$ to fuse with other sensor data to complete the final indoor activity recognition.

### 4.3. Motion Feature Extraction

We use acceleration $\left\{A_{x},A_{y},A_{z}\right\}$ and angular velocity $\left\{G_{x},G_{y},G_{z}\right\}$ obtained from acceleration sensors and gyroscopes for motion feature extraction. To extract the temporal feature information of these signals, we design a feature extraction network (MFN) based on transformer encoder structure [33]. The structure of MFN is shown in Figure 7. Here, our goal is to extract features from one-dimensional time-series sensor signals, so we use the encoder part of the transformer without using the input embedding and decoder part. The encoder stack consists of six layers and the number of attention head is eight. At the end of the encoder, we use a fully connected layer to obtain a 128-dimensional feature vector. In order to meet the input format of the transformer decoder, each 1×1024 sensor signal is divided into 16×64 in chronological order. This can be understood as treating the input sequence as text, with each input sample consisting of 16 words, each word having a dimension of 64. We add a fully connected layer at the end of the encoder to map the 16×64 output to a feature vector of 1×128 dimensions. With all the acceleration and angular velocity data, a total of six feature vectors $Z_{t}=\left\{Z_{a_{x}},Z_{a_{y}},Z_{a_{z}},Z_{g_{x}},Z_{g_{y}},Z_{g_{z}}\right\}$ are finally obtained.

### 4.4. Multimodal Feature Fusion and Indoor Activity Recognition

The structure of the multimodal feature fusion layer is shown in Figure 8. The data are acquired from the same sensor, such as $Z_{a_{x}},Z_{a_{y}}$, and $Z_{a_{z}}$, using feature concatenation. Thus, the motion feature vectors $Z_{a_{x}},Z_{a_{y}},Z_{a_{z}},Z_{g_{x}},Z_{g_{y}},Z_{g_{z}}$ are fused into vectors $Z_{a},Z_{g}$. Then, we fuse the feature vectors $Z_{a},Z_{g},Z_{l}$ from different sensors into a fused 512-dimension feature $Z_{m}$ using the idea of low-rank multimodal fusion (LMF) [34].

The fusion process essentially aims to first transform the input feature representations $Z_{a},Z_{g},Z_{l}$ into a high-dimensional tensor *Z*, and then map them back to a low-dimensional output vector $Z_{m}$ through a linear layer.
(4)$$Z=Z_{a}⊗Z_{g}⊗Z_{l}$$
(5)$$Z_{m}=W\cdot Z+b$$
where *W* is the weight of the linear layer, and *b* is the bias.

Assuming that the dimensions of the three input features are $d_{a},d_{g}$, and $d_{l}$, respectively, then the dimension of *Z* would be $d_{a}\times d_{g}\times d_{l}$. Directly fusing them in this way will greatly increase the computational complexity. To solve this problem, we can represent the weight tensor *W* in the following low-rank tensor way:(6)$$W=\sum_{i=1}^{r}w_{a}^{\left(r\right)}⊗w_{g}^{\left(r\right)}⊗w_{l}^{\left(r\right)}$$
where $w_{a}^{\left(r\right)},w_{g}^{\left(r\right)}$, and $w_{l}^{\left(r\right)}$ are the corresponding low-rank factors of the modality a,g, and *l*, and *r* is the fixed rank that we set as 4. We recommend readers to refer to [34] for more detailed information on low-rank factor decomposition. By using the low-rank decomposition instead of the original vector composition method, $Z_{m}$ can be directly obtained without calculating high-dimensional *Z*.
(7)$$\begin{array}{cc} Z_{m} & =(\sum_{i=1}^{r}w_{a}^{\left(r\right)}⊗w_{g}^{\left(r\right)}⊗w_{l}^{\left(r\right)})\cdot Z\\ \; & =\left(\sum_{i=1}^{r}w_{a}^{\left(r\right)}\cdot Z_{a}\right)\circ \left(\sum_{i=1}^{r}w_{g}^{\left(r\right)}\cdot Z_{g}\right)\circ \left(\sum_{i=1}^{r}w_{l}^{\left(r\right)}\cdot Z_{l}\right) \end{array}$$

The fused feature $Z_{m}$ is then sent to the indoor activity recognition subnetwork (IAN), which is basically a classifier and has a fully connected layer and a softmax layer.

## 5. Experiment

### 5.1. Experiment and Dataset Settings

As mentioned in the data acquisition section, a total of 160 sets of data sequences with room labels were collected, all of which contain geomagnetic data and ambient light information. Among them, 130 sets also have activity labels, which not only contain geomagnetic information and ambient light information, but also acceleration and angular velocity information. We divided all the data into sequence segments with a length of 1024, and the overlapping step size during segmentation was 200. A total of 2196 sets of sequence segments were obtained. All experiments were performed on a computer with an Nvidia RTX 3090 GPU using PyTorch as the deep learning framework.

#### 5.1.1. Training and Testing of LFN

To verify the effectiveness of the proposed location feature extraction subnetwork (LFN), we first train a indoor position classification network based on LFN proposed in this article. We keep the 6-d fully connected layer shown in Figure 6 and classify the input geomagnetic and ambient light data into one of the six room locations. We also train an LFN network without the ambient light data for comparison. Adam is chosen as the network’s optimizer, cross-entropy loss is the loss function, we train LFN with a batch size of 32 for 300 iterations, and the learning rate is set to 0.01.

In the training and testing process of LFN, all data were used. After data segmentation, we selected 80% of the data sequences as the training set and 20% as the test set. Table 2 shows the detailed information of the dataset used for LFN.

#### 5.1.2. Training and Testing of LM-IAR

After training the LFN, we trained the whole LM-IAR modal and tested its accuracy in indoor activity recognition. We removed the final fully connected layer of LFN, froze the remaining parameters, and integrated them into the overall LM-IAR model. We only used data with activity labels for further training to optimize the parameters of other parts of the model. The data information used in this period of experiment is shown in Table 3.

To verify the effectiveness of using location information to assist in indoor activity recognition, we conducted ablation experiments on our model. We compared our model with other activity recognition models, which were constructed based on LSTM and RNN, respectively. We further explored the impact of the sequence length and the different lifestyle habits of the participants on the test results.

### 5.2. Evaluation Criteria

We use the confusion matrix to demonstrate the recognition differences between different categories and use top-1 accuracy as the overall evaluation criteria for the model:(8)$$Top-1accuracy=\frac{\sum_{i=0}^{N-1}\left(f\left(x_{i}\right)==y_{i}\right)}{N}$$
where $x_{i}$ is the input data, $y_{i}$ is the label of $x_{i}$, the total number of classes of input data is *N*, and *f* denotes our model.

### 5.3. Experiment Results and Discussions

Figure 9 shows the location classification results with LFN. The top-1 accuracy of LFN is 90.50%. The experimental results indicate that, using our method, it is feasible to distinguish the user’s room using geomagnetic and ambient light signals collected from mobile phones. It can also be seen that in this experiment, the positioning accuracy using geomagnetic signals alone is much lower, with an overall top-1 accuracy of 77%, and the resolution of the living room and dining room is very low. This may be due to the fact that in the experimental environment of this article, the living room and dining room are more spacious compared to other rooms, and their geomagnetic variation characteristics are not obvious. From the confusion matrixes, we can see that after introducing ambient light information to assist in classification, the recognition accuracy of these two room has increased.

Table 4 shows the indoor activity recognition results by LM-IAR. To verify whether the idea of combining the location of activities with family activities effectively improves the efficiency of family activity recognition, we conducted the experiment with two variants of the model:LM-IAR-nl: We removed the light intensity data from the LFN.M-IAR: We removed the LFN part and only kept the motion feature extraction and feature fusion part.

Figure 10 shows the confusion matrix of LM-IAR and M-IAR. For M-IAR, the recognition rate of using the bathroom is the highest among the six activities, while the recognition rate of desk work is the lowest. Many desk works are mistakenly identified as eating and sleeping, and 21% of eating is also mistakenly identified as desk work, which is consistent with our previous analysis that the inertial data collected by mobile phone sensors can distinguish actions, but it is difficult to distinguish complex activities with similar actions such as eating and desk work. From Figure 10, we can see that introducing location features can help distinguish these two kinds of activities better, and the recognition accuracy reached 77% and 75%, respectively.

Table 5 compares our method with other human activity recognition methods. We implement a state-of-the-art LSTM-CNN model described in [20] and classic LSTM and RNN model described in [19]. It should be noted that these methods themselves are based on inertial sensor data to identify activities, without using positional information. Therefore, we introduce the variant M-IAR of our model into this comparative experiment as a fair comparison, as it also does not integrate location information. Furthermore, we combine the LFN network proposed in this article with several other methods, and we refer to these combined models as LFN-LSTM, LFN-RNN, and LFN-LSTM-CNN. Thanks to the relatively independent network structure of LFN, we are able to fuse the position features extracted by LFN with the action features extracted by other models. It should be noted that for three combined models, we only retrained the classifier after feature fusion without changing the parameters of LFN and their original motion feature abstraction networks.

The results show that after integrating location features, the recognition accuracy of the model for indoor activities is significantly improved, and among different motion feature extraction networks, our proposed network has the highest accuracy. We speculate that this is due to the long time span of the test data, making it difficult for RNN structural models to extract contextual information, which further affects their classification accuracy. However, the transformer-decoder-based network used in this article can more effectively process long sequence data.

We further explore the impact of data segmentation length on model accuracy and speed, and the test results are shown in Figure 11 and Figure 12. As shown in the figure, the longer the data, the higher the accuracy. This is because, on the one hand, a longer sequence of inertial sensor signals can better reflect motion characteristics, and on the other hand, a longer sequence of geomagnetic signals usually means covering a wider range of paths, which helps to more accurately identify positions. However, we can also observe that when the sequence length exceeds 1024, the rate of accuracy growth slows down. Considering that the processing time for longer sequences will be longer, we choose an input length of 1024 in the trade-off between accuracy and speed.

Figure 13 shows the relationship between the accuracy of indoor activity recognition and different experimental participants. Our dataset was collected by three volunteers. The mixed experiment involves uniformly mixing all data and dividing them into training and testing sets. Tests 1 to 3 were conducted only using data collected by volunteers 1–3 as the test set, respectively, while we trained the model with data collected by other volunteers. Train–test 1 to 3 used 40% of data from volunteers with corresponding numbers and all the data from other volunteers as training set, and used the remaining 60% of data from volunteers with corresponding numbers as the test set.

From this set of experiments, it can be seen that the method proposed in this paper is user-sensitive, and the testing accuracy is not very high when the testers have never participated in model training. This is because, unlike individual actions such as walking or standing, the indoor family activities to be identified in this experiment are more complex and have a greater relationship with the lifestyle habits of the participants. Therefore, it is necessary to perform transfer training and fine-tuning on the model for certain users before application.

## 6. Conclusions

Considering the difficulty of identifying complex indoor activities solely using inertial sensors, in this study, we proposed a method of extracting positional features from geomagnetic and ambient light signals collected by built-in sensors on mobile phones, and fusing them with motion features obtained by inertial sensors to achieve accurate recognition of indoor activities. We converted the geomagnetic sequence and ambient light sequence into images and designed a position feature extraction network based on 2D-CNN. Meanwhile, we used the transformer encoder structure to extract the temporal data features of the accelerometer and gyroscope. Then, we combined spatial and temporal features to enhance the saliency of activity features. We conducted data collection and experiments in an indoor environment containing six rooms, and trained a model to recognize several types of indoor activities, including eating, sleeping, going to the bathroom, doing laundry, cooking, and desk work. We compared our model with the method that only uses motion features, and the experimental results showed that our model can better distinguish various indoor activities compared to the model that only uses inertial sensors. Due to the good portability of our proposed position feature extraction network, we also combined it with other advanced methods, and experimental results showed that the position information can also improve the activity recognition accuracy of these methods.

## Figures and Tables

**Figure 1 sensors-24-03367-f001:**
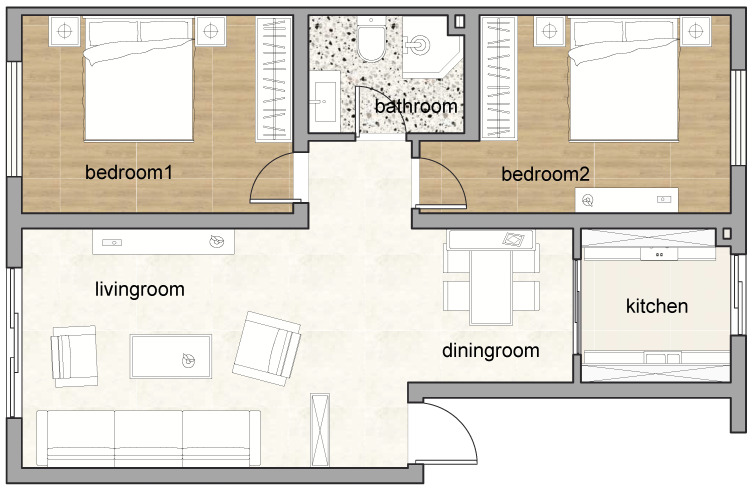
Apartment layout.

**Figure 2 sensors-24-03367-f002:**
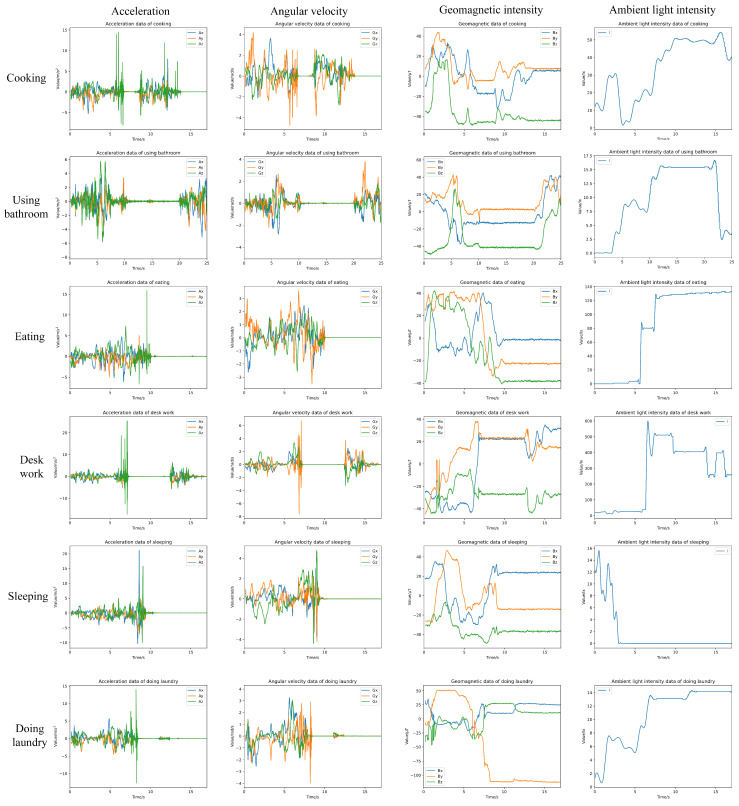
Sensor data collected under different activities.

**Figure 3 sensors-24-03367-f003:**
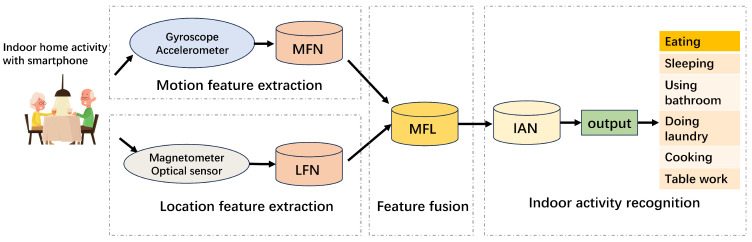
Overview of the LM-IAR workflow.

**Figure 4 sensors-24-03367-f004:**
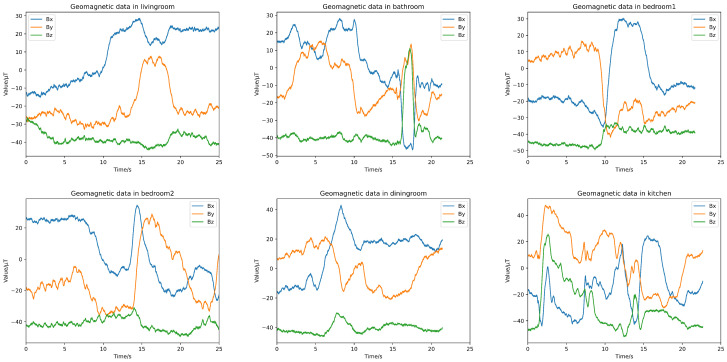
Geomagnetic data collected from different rooms.

**Figure 5 sensors-24-03367-f005:**
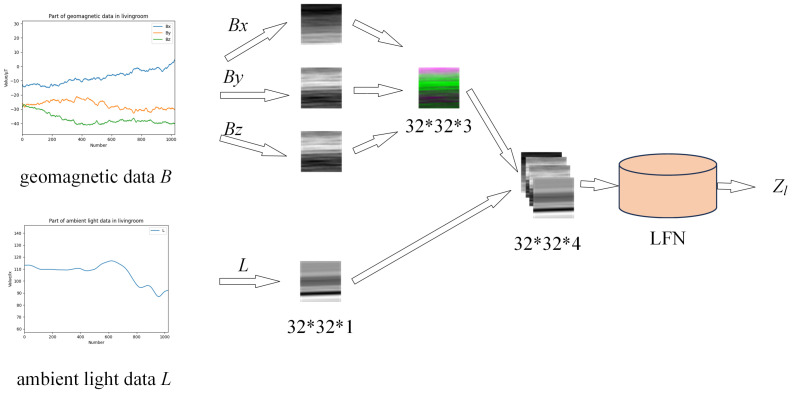
Data formation of LFN.

**Figure 6 sensors-24-03367-f006:**
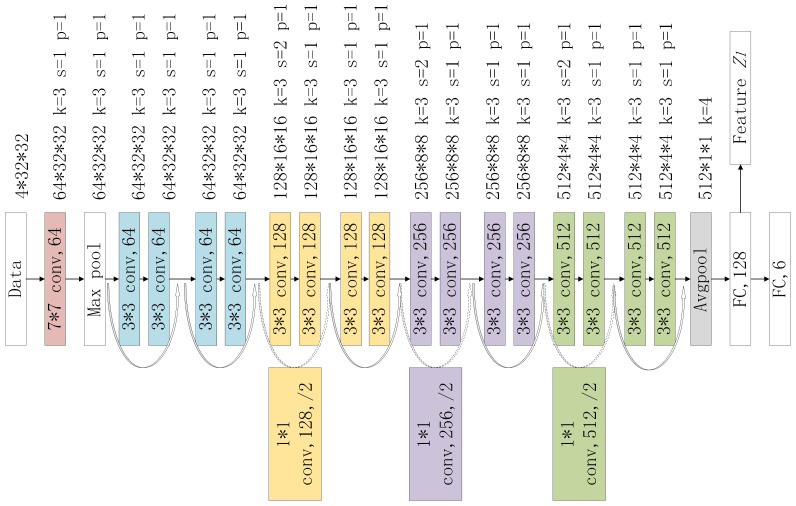
LFN structure.

**Figure 7 sensors-24-03367-f007:**
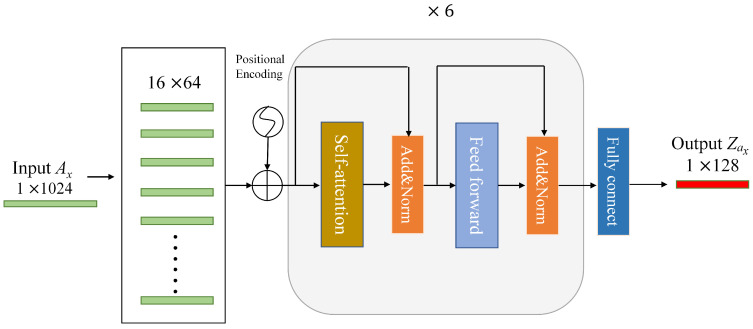
MFN structure.

**Figure 8 sensors-24-03367-f008:**
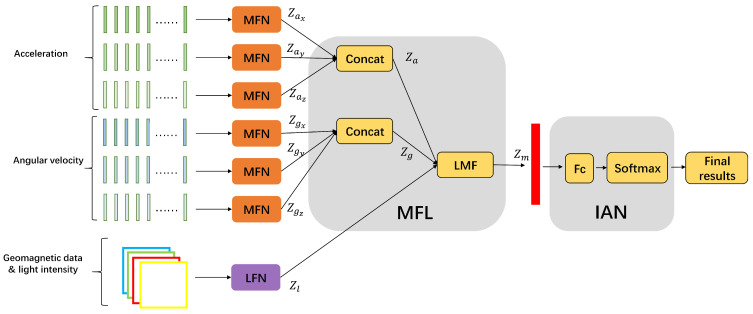
Modal fusion layer (MFL) structure.

**Figure 9 sensors-24-03367-f009:**
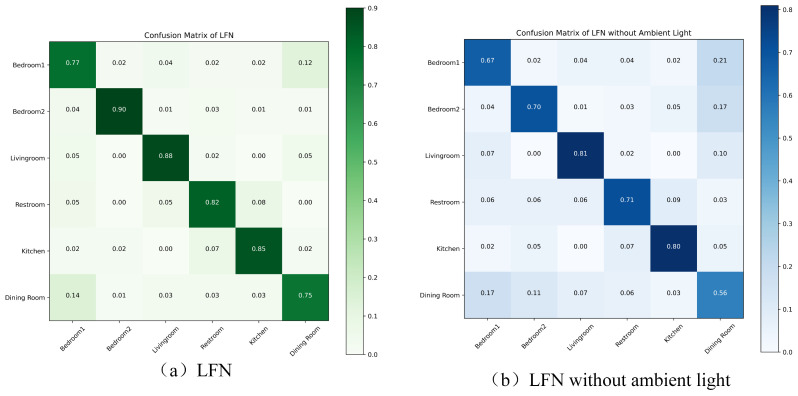
Confusion matrixes of LFN.

**Figure 10 sensors-24-03367-f010:**
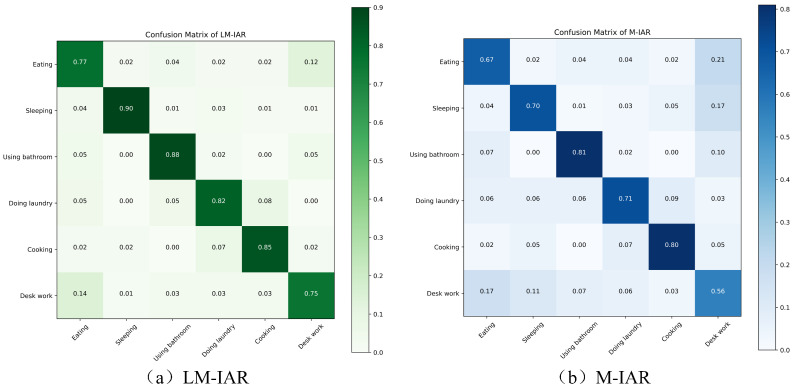
Confusion matrixes of LM-IAR and M-IAR.

**Figure 11 sensors-24-03367-f011:**
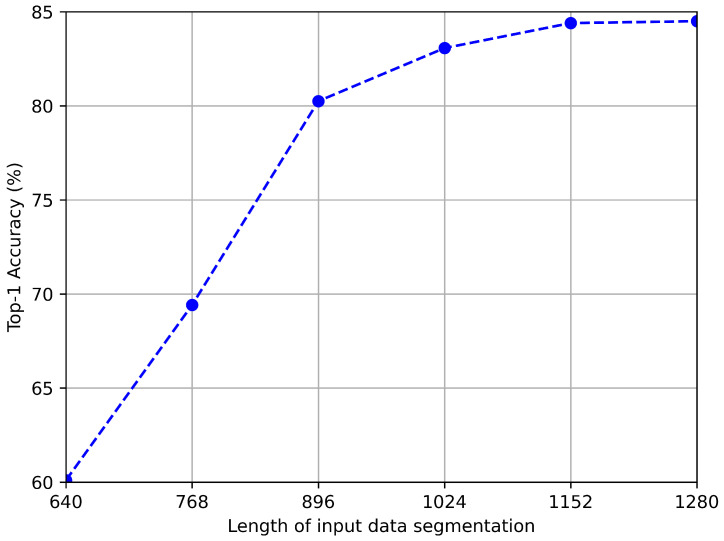
Recognition accuracy with different length of data segmentation.

**Figure 12 sensors-24-03367-f012:**
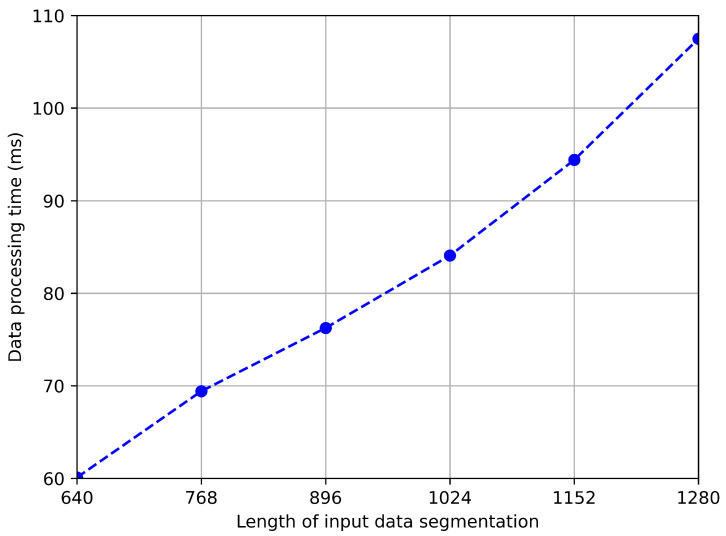
Inference time with different length of data segmentation.

**Figure 13 sensors-24-03367-f013:**
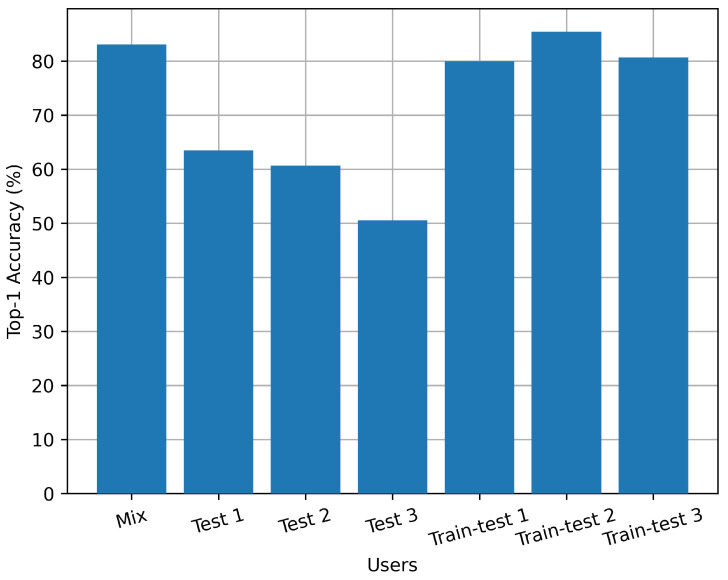
Accuracy with experiment participant.

**Table 1 sensors-24-03367-t001:** The total number of collected sets of sensor data sequences.

Activity	Bedroom1	Bedroom2	Livingroom	Restroom	Kitchen	Diningroom	Total (by Activity)
Eating	-	-	-	-	-	20	20
Sleeping	20	10	5	-	-	-	35
Using bathroom	-	-	-	15	-	-	15
Doing laundry	-	-	-	15	-	-	15
Cooking	-	-	-	-	15	-	15
Desk work	-	20	10	-	-	-	30
Walking (*B* and *L* only)	5	5	5	5	5	5	-
Total (by room)	25	35	20	35	20	25	-

**Table 2 sensors-24-03367-t002:** Number of sequence sets for LFN.

Room	Train	Test
Bedroom1	292	74
Bedroom2	350	89
Livingroom	274	68
Restroom	331	80
Kitchen	239	60
Diningroom	271	68
Total	1757	439

**Table 3 sensors-24-03367-t003:** Number of sequence sets for the LM-IAR.

Activity	Train	Test
Eating	202	48
Sleeping	362	80
Using Bathroom	150	42
Doing Laundry	143	38
Cooking	151	41
Desk Work	272	70
Total	1280	319

**Table 4 sensors-24-03367-t004:** Ablation experiment results of indoor activity recognition (%).

Method	Geomagnetic Intensity	Angular Velocity	Accelartion	Ambient Light Intensity	Top-1 Acc
LM-IAR	✓	✓	✓	✓	83.07
LM-IAR-nl	✓	✓	✓	-	71.19
M-IAR	-	✓	✓	-	69.42

**Table 5 sensors-24-03367-t005:** Experiment results comparison with other methods.

Method	Location Feature	Motion Feature	Top-1 Acc (%)
LSTM	-	✓	60.03
RNN	-	✓	53.19
LSTM-CNN	-	✓	67.33
M-IAR	-	✓	69.42
LFN-LSTM	✓	✓	77.11
LFN-RNN	✓	✓	74.92
LFN-LSTM-CNN	✓	✓	82.13
LM-IAR	✓	✓	83.07

## Data Availability

The data presented in this study are available on request from the corresponding author.

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
