# Peer review of "Position-Aware Indoor Human Activity Recognition Using Multisensors Embedded in Smartphones"

_sensors, 2024, doi:10.3390/s24113367_

Round 1

Reviewer 1 Report

Comments and Suggestions for Authors

This paper mainly proposed a human activity recognition method combining with indoor positioning based on the built-in sensors of mobile phone, which can improve the recognition performance by increasing the spatial and temporal correlation between activity types and lifestyle habits. Specifically, a location feature 

extraction subnetwork (LFN) using data from geomagnetic sensor and optical sensor, a motion feature extraction subnetwork (MFN) using gyroscopes and accelerometers signals, and a feature fusion layer that fuse multi-modal location features and acition features are designed. The experimental results demonstrated the feasibility and superority of this proposed method.

There are some comments as following.

1) Please add the signal samples of different human activities for each type of sensor.

2) For the different types of sensors, all the received signals are represented by one-dimensional time series. However, two different subnetworks are designed separately to extract location feature and motion feature. What is the motivation of designing two different subnetworks? Please provide the detailed explaination.

3) Please unify the units of time. In Figure 2 to Figure 5, the time is represented by seconds. However, in Figure 14, the times is denoted by the sequence length.

4) More evaluation metrics should be considered, such as network size and computation time.

5) The figures need to be redrawn to ensure better clarity.

6) Following the equations, the word 'Where' should be 'where'.

Author Response

Thank you for your decision and valuable comments on our manuscript. We agree with your suggestions and have carefully revised the manuscript based on them. Please see the attachment for our point-by-point response.

Reviewer 2 Report

Comments and Suggestions for Authors

In this paper, the authors propose the idea of integrating the location and activity information of a user to recognize his/her indoor activities based on the data measured by multiple sensors embedded in a smartphone. Through extensive experiments based on the real data measured by volunteers at an apartment, they demonstrate that the performance of the LM-IAR model, which consists of the four major blocks of motion feature extraction, location feature extraction, feature fusion, and indoor activity recognition, outperforms other models solely based on motion features in recognizing six predefined activities.

The overall presentation, including that of the proposed LM-IAR model and the experimental results with lots of figures and tables in sections 4 and 5, is good. Also, and the quality of English is quite satisfactory.

From the reviewer's point of view, however, the major issue of the current work is the lack of the justification of the use of smartphone data for motion features. 

Though the authors argue against the use of wearable devices as data sources in favour of smartphones, they do not provide compelling rationales for the use of smartphones in achieving activity recognition. For instance, of the six activities selected for this work, it is hard to imagine that people carry smartphones while eating, doing laundry, and cooking. Typically, they put smartphones on nearby tables or kitchen sink, or, at best, they put smartphones inside their pockets while doing those activities; for the former, the extraction of useful motion features from smartphone data is impossible, while, for the latter, the extraction of motion features is possible but the optical sensors cannot provide useful information for location estimation.

Related with the major issue above, the details of how volunteers carried their smartphones should have been discussed in Section 3 but not, which would significantly affect the experimental results presented in Section 5. Maybe this information (i.e., how to carry smartphones during the data acquisition) is closely related with the observation based on Fig. 15 in section 5 regarding the sensitivity of the proposed method to users.

In fact, the experiments of the current work are designed in such a way that knowing the location information alone gives enough information on the activities; for example, Table 1 shows that we can easily identify the activities solely based on the location information in the cases of Bedroom1 (i.e., Sleeping), Kitchen (i.e., Cooking), and Diningroom (i.e., Eating); in the remaining cases of Bedroom2, Livingroom, and Restroom, we can narrow down the number of possible activities into just two. In view of this, the outstanding performance of the proposed model in comparison to other methods based only on motion features, which is summarized in Table 5, is quite expected; it is likely that a much simpler model (e.g., classical feed-forward neural networks) than the proposed LM-IAR also outperforms any other models based only on motion features because much of useful information for activity recognition is provided by location features.

In this regard, therefore, the authors are recommended to provide the said justification of the use of smartphone data for motion features and the details of how volunteers carried their smartphones during the data acquisition. It would be also interesting to include additional methods (even the authors' own ones for self comparison) based on both location and motion features in Table 5.

Minor comment:

- Page 5, line 191: "... collected rom different sensors ..." -> "... collected from different sensors ..."

Author Response

(The authors gave the same response as above.)

Reviewer 3 Report

Comments and Suggestions for Authors

The Introduction starts with a description of human activity recognition (HAR) and its relevance in elderly care and rehabilitation, the limitations of existing research (simple activities) is pointed out and the coordinated approach presented in this paper is motivated. The challenges related to designing such solutions are pointed out and the approach based on smartphones is introduced and benefits are emphasized (such as not having to buy additional hardware, familiarity, and user friendliness).  The different sensors in a smartphone are enumerated and their relevance and use in the approach is described. Six different indoor activities are the basis of the experiment carried out to evaluate the approach.  

In the related work section different approaches of activity recognition, video/vision-based methods, and sensor paste methods with different advantages and disadvantages are pointed out, together with classification algorithms which are widely used, for example K-nearest neighbor and Bayesian approaches, neural networks such as LSTM, deep learning approaches and geomagnetism.  

In section 3, data acquisition methods are described on the basis of a figure of the floor plan of the model apartment, where the experiments toook place (including two bedrooms, a kitchen, bathroom, a living room, and a dining room). The tasks which were planned to be recognized are, for example eating, which takes place sitting on a desk. The raw data was recorded with a sensor logger app on Android based smartphones, table 1 shows an overview on the numbers of the collected data connected to the different rooms and the different activities, figures two and three are show different data from geomagnetic sensors and acceleration data in the living room, also figures 4 and 5 show different curves of data, formulae are provided to show the basis mathematic formulae used for data processing. 

In section 4, proposed method details of the applied activity recognition and positioning recognition strategies are described, which are based on the fusion of different sensors in a smartphone (e.g. gyroscope, accelerometer) and the recognition/estimation of activity. The experiment carried out is described in section 5 and consisted of the collection of 160 sets of data sequences in different quality. Additionally, the approach for training and testing the applied algorithms is described (run on an Nvidia GPU Hardware with PyTorch as Deepl Learning Framework) .  

In the discussion section, the accuracy of the presented approach (90,5%) compared to other approaches is emphasized, and the performance of an additional mixed experiment is described, which resulted in low activity recognition accuracy (related to individual differences of participants. The paper finished with conclusions, where the authors emphasize the advantages of their own approach 

Review: 

The paper addresses an important topic, the introduction and motivation for the approach is clear, also the emphasis on the challenges and limitations of existing approaches. The related work is comprehensive, and the authors appropriately investigated the related literature. The followed approach is clearly presented, and the chosen methods of data collection data manipulation and analysis are accurately described. That conclusion is quite short and more or less repeats the motivation of the approach followed.  

I have some doubts regarding the followed approach, specifically the pointed-out advantages to use a smartphone instead of other sensors. Although I would agree that the installation of additional hardware has disadvantages, basing activity recognition in living circumstances on smartphones might also have limitations. At least in some cultures and certain shares of population, smartphones are not always carried by their owners, specifically when they are at home. The phones are, for example, charged or put in certain places until their owners leave their home. In these cases, utilizing the phone’s sensors may not be possible in real life circumstances. Another limitation of the approach is, for example, that even smart meters (based on non-intrusive load monitoring – NILM) could identify activity, when it is related to the use of a device without having to track the specific position/location. Because, as the authors point out themselves, cooking is done in the kitchen, and the On-status of the kitchen stove would be sufficient to identify this activity. Combined with other data such as time of the day, weekday/weekend it would be possible to extract information of the same quality as the authors propose on the basis of a smartphone. 

In regard to the presentation and formatting of the paper, several typos, word repetitions, and incomplete formulations can be found. For example, line 6 in the abstract: “And We establishe; In the introduction, line 23focuse; line 27, should read “it require with an “s”; page 2, line 66 “we delevope”, to mention only a few.  Some of the figures use much space but do not convey much relevant information (e.g. Figure 3, figure 4). They could possibly be combined / reduced in size.

Comments on the Quality of English Language

has to be improved, see above.

Author Response

(The authors gave the same response as above.)

Round 2

Reviewer 1 Report

Comments and Suggestions for Authors

The authors have addressed all my concerns, so I think this manuscript can be accepted.

Reviewer 2 Report

Comments and Suggestions for Authors

The revised version, together with the authors' response, successfully addresses the issues raised during the 1st round of the review.

Reviewer 3 Report

Comments and Suggestions for Authors

The authors did appropriately consider my comments and revised the paper thoroughly. However, I'm still not convinced by the research approach because of the limitations of smartphones in the HAR context I pointed out in my first comments. Therefore, my overall rating is still "average". Still, I have no objections to the paper being published.